# High prevalence of extended-spectrum beta-lactamase-producing *Escherichia coli* and *Klebsiella pneumoniae* fecal carriage among children under five years in Addis Ababa, Ethiopia

**Mekdes Alemu Tola**[1]*, **Negga Asamene Abera**[2], **Yonas Mekonnen Gebeyehu**[2], **Surafel Fentaw Dinku**[2], **Kassu Desta Tullu**[3]

1 Armauer Hansen Research Institute, Addis Ababa, Ethiopia, 2 Department of Clinical Bacteriology and Mycology, National Reference Laboratory, Ethiopian Public Health Institute, Addis Ababa, Ethiopia, 3 Department of Medical Laboratory Sciences, College of Health Sciences, Addis Ababa University, Addis Ababa, Ethiopia

* mekdesalemu1@gmail.com

**Data Availability Statement:** The data used to support the findings of this study are included in this manuscript.

## Abstract

### Background

Extended-spectrum beta-lactamase (ESBL) producing bacteria present an ever-growing burden in the hospital and community settings. Data on the prevalence of ESBL fecal carriage remain scarce in Ethiopia. Therefore, this study aimed to determine the prevalence of ESBL producing *Escherichia coli* and *Klebsiella pneumoniae* fecal carriage among children under five years in Addis Ababa, Ethiopia.

### Methods

A facility-based cross-sectional study was conducted from April to May 2017. A total of 269 fecal/rectal swab samples were cultured on MacConkey agar. All positive cultures were characterized by colony morphology, Gram stain, and standard biochemical tests. Further, bacteria identification, antimicrobial susceptibility testing, and phenotypic detection of ESBL production were performed using VITEK 2 Compact as per the instruction of the manufacturer. Socio-demographic and risk factors data were collected using questionnaires. Data were entered by EPI INFO version 7.2.1.0 and analyzed by SPSS version 20.

### Results

The overall prevalence of ESBL-producing *E. coli* and *K. pneumoniae* was 17.1% (46/269; 95% CI: 12.9%–22.7%). A total of 47 isolates were ESBL-positive, of which, 83.0% were *E. coli* and 17.0% were *K. pneumoniae*. ESBL producing *E. coli* and *K. pneumoniae* isolates were also showed high levels of MDR (93.6%) and high rates of co-resistance to aminoglycosides, fluoroquinolones, and trimethoprim-sulfamethoxazole. However, all isolates were carbapenem susceptible. In the risk factors analysis, Children's mothers who had lower

**Funding:** The author(s) received no specific funding for this work.

**Competing interests:** The authors have declared that no competing interests exist.

educational level (primary school) (OR: 2.472, 95% CI: 1.323–4.618, P = 0.0062) and children who used tap water for drinking (OR: 1.714, 95% CI: 1.001–3.659, P = 0.048) were found to be significantly associated with higher ESBL fecal carriage.

## Conclusions

In this study, the high prevalence rate of ESBL producing *E. coli* and *K. pneumoniae* fecal carriage and high level of multidrug resistance among ESBL producing *E. coli* and *K. pneumoniae* were demonstrated. This suggested that the necessity of routine screening of ESBL is crucial for the early detection and appropriate antibiotics selection for infection caused by ESBL producing pathogens.

## Introduction

Antimicrobial resistance among bacterial strains is an emerging problem worldwide [1] with serious consequences on the treatment of infectious diseases [2]. Beta-lactam drugs like penicillins, cephalosporins, carbapenems, and aztreonam are common antibiotics used to combat most bacterial infections [3]. Indiscriminate use of third-generation cephalosporins to treat gram-negative bacterial infections is partly responsible for the emergence of resistance to beta-lactam antibiotics [4], which subsequently led to the emergence of Extended Spectrum Beta-Lactamases (ESBL) producing organisms [3].

ESBL are enzymes produced by gram-negative bacteria that mediate resistance to penicillins, cephalosporins, and monobactams [5]. There is no consensus on the precise definition of ESBL. A commonly used working definition is that the ESBL are β-lactamases capable of conferring bacterial resistance to the penicillins, first, second, and third-generation cephalosporins, and aztreonam (but not the cephamycins or carbapenems) by hydrolysis of these antibiotics, and which are inhibited by β-lactamase inhibitors such as clavulanic acid [6, 7].

ESBLs recognized in the 1980s in *Klebsiella species* and later in *Escherichia coli* and other gram-negative bacilli are currently spreading rapidly amongst other members of *Enterobacteriaceae*, largely due to genes located on plasmids that can distribute across species barriers [8]. The majority ESBLs can be divided into three genotypes: TEM, SHV, and CTX-M [9–11]. TEM and SHV-type β-lactamases, mainly produced by *K. pneumoniae*, have spread throughout hospital settings, and CTX-M enzymes, mainly produced by *E. coli*, have become predominant in the community [12].

*Ecoli* and *K. pneumoniae* are common species of *Enterobacteriaceae* that both have pathogenic potential and that frequently incorporate ESBL-encoding genes. The Infectious Diseases Society of America has listed them as two out of six pathogens for which new drugs are urgently needed to combat resistance development [13]. *E. coli* and *K. pneumoniae* cause various infections, such as urinary tract infections (UTIs), gastroenteritis, infant meningitis, wound infection, pneumonia, peritonitis, and bacteremia in both nosocomial and community settings [14–16].

ESBL was initially associated with nosocomial outbreaks caused by single enzyme-producing strains, but recent studies have revealed the existence of more complex situations, with significant increases in the frequency of community isolates [17]. ESBL determinants have been detected not only in clinical isolates but also in commensal bacteria from humans and animals, and isolates from products of the food chain and sewage, revealing distribution and suggesting the presence of environmental reservoirs for these resistance determinants [18].

A threatening epidemiological problem is the dissemination of ESBL-producing organisms to healthy people in the community, which might depend on the frequency of ESBL fecal carriage as well as on the presence of ESBL-producing organisms in the food chain [19]. The digestive tract is the main reservoir from which *Enterobacteria*ceae originate, whatever the type (community or hospital-acquired) of infection. It is also a melting pot where exchanges of resistance genes occur and antibiotic treatments select for the overgrowth of resistant bacteria [20]. Colonization in the intestinal compartment by ESBL-producing isolates has been associated with a high risk for developing infection due to ESBL producers [21]. It may also serve as a reservoir for ESBL resistance genes that can undergo horizontal transmission to other *Enterobacteriaceae* [22].

Multidrug resistance has been increased all over the world that is considered a public health threat. Several recent investigations reported the emergence of multidrug-resistant bacterial pathogens from different origins including humans, poultry, cattle, and fish that increase the need for routine application of the antimicrobial susceptibility testing to detect the antibiotic of choice as well as the screening of the emerging MDR strains [23–30].

Evidence-based data is useful to implement an antimicrobial drug policy and to adopt best-practice infection control measures to prevent the spread of those organisms in health care facilities as well as in the community. However, data on the prevalence of ESBL fecal carriage remain scarce in Ethiopia and to the best of our knowledge, there was no study focusing on children particularly in community settings. Therefore, this study aimed to determine the prevalence of ESBL producing *E. coli* and *K. pneumoniae* fecal carriage among children under five years in Addis Ababa, Ethiopia.

## Materials and methods

### Study design, study setting, and data collection

A facility-based cross-sectional study was conducted from April to May 2017 at Addis Raey public health center, Addis Ababa, Ethiopia. A consecutive sampling technique was used to collect the stool/rectal specimens from under five years children attending outpatient department (OPD) service after verbal informed consent was taken from all parents or guardians on the behalf of the children. Information on socio-demographic and associated risk factors was collected using structured pretested questionnaires. Data concerning previous antibiotics usage (for the last 12 months) was obtained through medical records review. Under-five years children received antibiotic treatment for the last one week (7 days) before data collection time and were unable to consent to participate in this study were the exclusion criteria. The laboratory analysis was performed at the Ethiopian Public Health Institute (EPHI) Clinical Bacteriology and Mycology National Reference Laboratory in Addis Ababa, Ethiopia.

### Laboratory investigation

**Specimen collection and processing.**   Fresh fecal/rectal swab specimens from 269 children were collected and put into a Cary-Blair transport medium, then stored in an ice pack and transported to Ethiopian Public Health Institute (EPHI) Clinical Bacteriology and Mycology National Reference Laboratory within the same day of collection. Fecal samples were inoculated on MacConkey agar plates and incubated at 37˚C for 24 hours on the same day of collection.

**Bacterial isolation and identification.**   All fecal/rectal swab samples were cultured on MacConkey agar (Oxoid Ltd., Basingstoke, United Kingdom) and incubated at 37˚C for 24 hours. *E. coli* and *K. pneumoniae* were isolated based on their colony morphology, pigment production (pink to colorless flat or mucoid colonies due to lactose fermentation), and Gram-

staining reaction (Gram-negative rods). Identification of isolates was confirmed by biochemical tests using standard procedures [31]. An isolate was considered as *E. coli* when it is Indole positive, citrate negative, lysine positive, gas and acid producer, ferments mannitol, urea negative, and motile. An isolate was considered as *K. pneumoniae* when it is indole negative, citrate positive, ferments mannitol, lysine positive, urea slows producing, and non-motile. Further bacterial identification, antibiotic susceptibility test (AST), and phenotypic detection of ESBL production were performed using VITEK 2 Compact (bioMe´rieux, France) as per the instruction of the manufacturer.

**Identification of bacterial isolates by VITEK 2.** The VITEK 2 compact system is an automated microbiology bacterial identification and antimicrobial susceptibility system. Uses advanced colorimetry technology to determine individual biochemical reactions contained in a variety of microbe identification cards. After inoculation with a standardized suspension of the unknown organism, each self-contained card is incubated and read by the instrument's internal optics. Comparison of results to known species-specific reactions in the VITEK 2 database yields organism identifications. A transmittance optical system allows the interpretation of test reactions using different wavelengths in the visible spectrum. During incubation, each test reaction is read every 15 minutes to measure either turbidity or colored products of substrate metabolism. In addition, a special algorithm is used to eliminate false readings due to small bubbles that may be present.

Gram-negative bacteria identification card (ID-GN) was used for the identification of isolated bacteria. This card is used for the automated identification of 135 taxa of the most significant fermenting and non-fermenting gram-negative bacilli.

Gram-negative bacteria identification card (ID-GN) (bioMe´rieux SA, France) was used for the identification of isolated bacteria. This card is used for the automated identification of 135 taxa of the most significant fermenting and non-fermenting gram-negative bacilli. The Bacterial suspension was prepared aseptically by transferring 3.0 ml of sterile saline (0.45% to 0.5% NaCl, pH 4.5 to 7.0) into two clear plastic (polystyrene) test tubes. Then using a sterile loop transfer a sufficient number of morphologically similar colonies (pure culture) to the first saline tubes. Check the density of suspension (equivalent to 0.50 to 0.63 McFarland standard using a turbidity meter (DensiChek). Transfer 145ul of the suspension prepared in the first tube to the second tube. The first bacterial suspension tube was used for identification and the second tube was used for AST and ESBL tests.

**Antibiotic susceptibility testing.** AST-GN86 card (bioMe´rieux SA, France) was used for AST and ESBL tests. Susceptibility test for 18 antimicrobials of different classes which includes: penicillins (ampicillin), beta-lactamase inhibitor combinations (amoxicillin/clavulanic acid and ampicillin/sulbactam), cephalosporins (cefazolin, cefuroxime, cefuroxime axetil, ceftazidime, ceftriaxone, and cefepime), carbapenems (ertapenem and imipenem), aminoglycosides (gentamicin and tobramycin), fluoroquinolones (ciprofloxacin and levofloxacin), tetracycline, nitrofurantoin, and trimethoprim/sulfamethoxazole was performed using AST-GN86 card by VITEK 2.

The results of the susceptibility test were interpreted as sensitive, intermediate, and resistant based on the Clinical and Laboratory Standards Institute (CLSI) guideline [32]. VITEK 2 also performs the Minimum Inhibitory Concentration (MIC). Multidrug resistance is defined as non-susceptibility to at least one agent in three or more antimicrobial categories [33].

**ESBL test.** VITEK 2 ESBL test is a confirmatory test to detect the presence of extended-spectrum beta-lactamase (ESBLs) in *E. coli*, *K. pneumoniae*, and *Klebsiella oxytoca*. AST-GN86 cards have drugs for ESBL confirmation.

Concentrations:

Cefepime (1 μg/ml)              -Cefepime/Clavulanic Acid (1/10 μg/ml)

Cefotaxime (0.5 μg/ml)                    -Cefotaxime/Clavulanic Acid (0.5/4 μg/ml)

Ceftazidime (0.5 μg/ml)                    -Ceftazidime/Clavulanic Acid (0.5/4 μg/ml)

The ESBL analysis for the VITEK 2 system is based on monitoring organism activity (growth) in seven different wells on the test card. One well is a control containing only growth media. The other six are cefepime, cefotaxime, and ceftazidime, each with and without clavulanic acid.

Organism activity is monitored in the control well to determine whether sufficient activity is present to complete the analysis and to determine the length of incubation. No ESBL result is reported unless the organism reaches predetermined growth thresholds. Once the organism reaches the exponential phase, incubation is extended a set amount of time to evaluate the activity in the antimicrobial wells with and without clavulanic acid.

Test Principle: The detection of an ESBL is based on the inhibition of activity in the presence of clavulanic acid. The VITEK 2 analysis looks for growth patterns that exhibit activity in the well containing the antimicrobial without clavulanic acid and limited activity in the corresponding antimicrobial well containing clavulanic acid. Each of the three pairs of wells is evaluated independently. If any one of the three pairs demonstrates the expected growth pattern (difference in activity with and without clavulanic acid) a positive test result is reported.

Result interpretation: Negative: strain does not produce ESBLs.

Positive: strain produces ESBL. Test interpretation for ESBL positive should be reported as resistant for all penicillins, cephalosporins, and aztreonam.

**Quality control.**   Quality control was done for all reagents. Culture media were tested for sterility and performance. Moreover, for the ESBL confirmatory test, standard organisms; *E. coli* ATCC 25922(ESBL negative) and *K. pneumoniae* ATCC 700603(ESBL positive) were performed as recommended by the CLSI guideline [32].

## Ethics approval and consent to participate

The study proposal was reviewed and approved by the department of research and ethics review committee of the Medical Laboratory Sciences, College of Health Sciences, Addis Ababa University (Ref. No. MLS/271/17). Written informed consent was obtained from each parent or guardian on behalf of the children after a brief explanation of the purpose of the study before the interview and sample collection.

## Statistical analysis

Data were entered using EPI INFO version 7.2.1.0 and analyzed using Statistical Package for Social Science (SPSS) version 20 (IBM-SPSS Inc., Chicago, IL, USA). A simple frequency was used to describe the study population with the socio-demographic, clinical condition, and other relevant variables. Chi-square analysis was used to explore risk factors associated with the prevalence of ESBL carriage by using an odds ratio with a 95% confidence limit. A value of $p < 0.05$ was considered to be statistically significant. The data were presented in tables.

## Results

### Socio-demographic characteristics

The characteristics of the patients are shown in Table 1. In total, 269 children were enrolled in this study. Of which, 139 (51.7%) were female and 130 (48.3%) were male. The median age of the study participants was 18 months (range from 1 to 59 months), 155 (57.6%) were infants (1–23 months) and 114 (42.4%) were children (24–59 months). Most of the study participants 206 (76.6%) had family members of 2 to 5. The majority of children's mothers 143 (53.2%) were within the age group of 25–34. The study participant's mother's educational status

**Table 1. Sociodemographic characteristics of study participants.**

| Characteristics | | | Frequency | Percent |
|---|---|---|---|---|
| Age of children | | 29days-23months | 155 | 57.6 |
| | | 24–59 months | 114 | 42.4 |
| Gender | | Male | 130 | 48.3 |
| | | Female | 139 | 51.7 |
| Family size | | 2–5 | 206 | 76.6 |
| | | 6–9 | 57 | 21.2 |
| | | 10–13 | 6 | 2.2 |
| Age of mothers | | 15–24 | 80 | 29.7 |
| | | 25–34 | 143 | 53.2 |
| | | 35–44 | 30 | 11.2 |
| | | 45–54 | 1 | 0.4 |
| | | Don't know | 15 | 5.6 |
| Mother Educational level | | Illiterate/cannot read and write/ | 53 | 19.7 |
| | | Illiterate /able to read and write/ | 3 | 1.1 |
| | | Primary | 130 | 48.3 |
| | | Secondary | 61 | 22.7 |
| | | College Graduate | 10 | 3.7 |
| | | Don't know | 12 | 4.5 |
| Parent Income | | <500 birr | 12 | 4.5 |
| | | 500-1000birr | 46 | 17.1 |
| | | 1000 -2000birr | 82 | 30.5 |
| | | >2000birr | 129 | 48.0 |
| Place of resident | | Addis Ababa | 260 | 96.7 |
| | | Out of Addis Ababa | 9 | 3.3 |
| House condition | | Private | 47 | 17.5 |
| | | Government rental | 87 | 32.3 |
| | | Private rental | 135 | 50.2 |
| Source of drinking water for the child | Tap Water | Yes | 128 | 47.6 |
| | | No | 141 | 52.4 |
| | Boiled & Cooled Water | Yes | 26 | 9.7 |
| | | No | 243 | 90.3 |
| | Treated Water | Yes | 34 | 12.6 |
| | | No | 235 | 87.4 |
| | Bottled Water | Yes | 113 | 42.0 |
| | | No | 156 | 58.0 |
| | Filtered Water | Yes | 1 | 0.4 |
| | | No | 268 | 99.6 |
| Toilet use for family | | Private | 33 | 12.3 |
| | | Communal | 236 | 87.7 |

showed that 130 (48.3%) of them attended primary school. Concerning the source of drinking water for the children, the majority were used tap water 128 (47.6%). The majority of children's family 236 (87.7%) were using communal latrines (Table 1).

## Clinical conditions of study participants

The nutritional status of most children indicated that normal 252 (93.6%). The majority of children were born in health facilities (health center 145 (53.9%) and hospital 108 (40.1%)).

About 220 (81.8%) of the study participants were born by normal delivery (vaginal delivery). Most children 216 (80.3%) had exposure to previous antibiotics usage. However, the majority of the study participants had no exposure to hospital visits 220 (81.8%), previous hospital admission 248 (92.2%), and previous surgery 267 (99.3%) (Table 2).

## Prevalence of *E. coli* and *Klebsiella* species

A total of 264 *E. coli* and *Klebsiella* species were isolated from 269 participants' fecal/rectal swab samples. E. *coli* 224 (84.8%) was the most commonly isolated bacteria followed by *K. pneumoniae* 39 (14.8%) and *K. oxytoca* 1 (0.4%).

## Prevalence of ESBL fecal carriage

The overall prevalence of ESBL producing *E. coli* and *K. pneumonia* fecal carriage among under five years children was 17.1% (46/269; 95% CI: 12.9%–22.7%). Out of 264 *E. coli* and *Klebsiella* species isolates, 47 were ESBL-positive. Of which, 83.0% (39/47) were *E. coli* and 17.0% (8/47) were *K. pneumoniae*. One child was ESBL-positive for both *E. coli* and *K. pneumoniae*. ESBL fecal carriage proportion was higher among children in the age groups of 29 days to 23 months 19.4% (30/155). There was no significant difference in the number of ESBL fecal carriage between males 17.7% (23/130) and females 16.5% (23/139). ESBL fecal carriage

**Table 2. Clinical conditions of study participants.**

| Characteristics | | Frequency | Percent |
|---|---|---|---|
| Nutrition Status | Normal | 252 | 93.6 |
| | MAM | 14 | 5.2 |
| | SAM | 3 | 1.1 |
| Place of birth | Home | 12 | 4.5 |
| | Health center | 145 | 53.9 |
| | Hospital | 108 | 40.1 |
| | Private Clinic | 4 | 1.5 |
| Mode of delivery | Vaginal Delivery | 220 | 81.8 |
| | Cesarean Section | 49 | 12.2 |
| Prior intake of antibiotics | Yes | 216 | 80.3 |
| | No | 53 | 19.7 |
| Hospital visit | Yes | 49 | 18.2 |
| | No | 220 | 81.8 |
| Previous hospital admission | Yes | 21 | 7.8 |
| | No | 248 | 92.2 |
| Previous surgery | Yes | 2 | 0.7 |
| | No | 267 | 99.3 |
| Diarrhea for the last three months | Yes | 130 | 48.3 |
| | No | 139 | 51.7 |
| GI symptom | Yes | 120 | 44.6 |
| | No | 149 | 55.4 |
| Number of visits | First | 56 | 20.8 |
| | Second | 63 | 23.4 |
| | Third | 87 | 32.3 |
| | More than three | 63 | 23.4 |

MAM: Moderate Acute Malnutrition SAM: Severe Acute Malnutrition GI: Gastroenteritis.

**Table 3. Fecal carriage of ESBL producing *E. coli* and *K. pneumoniae* among under five years children.**

| Characteristics | | Total | ESBL | |
|---|---|---|---|---|
| | | N = 269 | Negative N = 223 | Positive N = 46 |
| | | | N (%) | N (%) |
| Age | 29days-23months | 155 | 125(80.6) | 30(19.4) |
| | 24–59 months | 114 | 98(86.0) | 16(14.0) |
| Sex | Male | 130 | 107(82.3) | 23(17.7) |
| | Female | 139 | 116(83.5) | 23(16.5) |
| Nutrition Status | Normal | 252 | 209(82.9) | 43(17.1) |
| | MAM | 14 | 11(78.6) | 3(21.4) |
| | SAM | 3 | 3(100) | 0(0.0) |
| Place of birth | Home | 12 | 9(75.0) | 3(25.0) |
| | Health center | 145 | 121(83.4) | 24(16.6) |
| | Hospital | 108 | 90(83.3) | 18(16.7) |
| | Private Clinic | 4 | 3(75.0) | 1(25.0) |
| Mode of delivery | Vaginal delivery | 220 | 182(82.7) | 38(17.3) |
| | Cesarean Section | 49 | 41(83.7) | 8(16.3) |
| Prior intake of antibiotics | Yes | 216 | 180(83.3) | 36(16.7) |
| | No | 53 | 43(81.1) | 10(18.9) |
| Hospital visit | Yes | 49 | 42(85.7) | 7(14.3) |
| | No | 220 | 181(82.3) | 39(17.7) |
| Previous hospital admission | Yes | 21 | 18(85.7) | 3(14.3) |
| | No | 248 | 205(82.7) | 43(17.3) |
| Previous surgery | Yes | 2 | 2(100.0) | 0(0.0) |
| | No | 267 | 221(82.8) | 46(17.2) |
| Diarrhea for the last three months | Yes | 130 | 108(83.1) | 22(16.9) |
| | No | 139 | 115(82.7) | 24(17.3) |
| GI symptom | Yes | 120 | 101(84.2) | 19(15.8) |
| | No | 149 | 122(81.9) | 27(18.1) |
| Number of visits | First | 56 | 45(80.4) | 11(19.6) |
| | Second | 63 | 46(73.0) | 17(27.0) |
| | Third | 87 | 78(89.7) | 9(10.3) |
| | More than three | 63 | 54(85.7) | 9(14.3) |

was slightly higher among children who had moderate acute malnutrition 21.4% (3/14). ESBL carriage was low in children with a history of prior intake of antibiotics 16.7% (36/216), the previous hospital visits 14.3% (7/49), hospital admission 14.3% (3/21), who had at least one episode of diarrhea for the last three months 16.9% (22/130) and GI symptoms 15.8% (19/120) than their counterparts (Table 3).

## Antimicrobial susceptibility pattern

The antibiotics susceptibility pattern of ESBL producing and non-ESBL producing *E. coli* and *K. pneumoniae* isolates is presented in Table 4. Overall, the highest resistance level was recorded for Ampicillin (77.2%), sulfamethoxazole-trimethoprim (60.8%), and tetracycline (57.4%). *K. pneumoniae* was 100% resistant to ampicillin in both ESBL and non-ESBL producers. A high level of resistance was observed against cephalosporins among ESBL producing *E. coli* and *K. pneumoniae* isolates compared to the non-ESBL producer that shows low or no resistance.

High level of resistance for trimethoprim/sulfamethoxazole (78.7%), followed by tetracycline (70.2%), ciprofloxacin (25.5%), and gentamicin (17%) were detected in ESBL producing

**Table 4. Antimicrobial resistance pattern of ESBL producing and non-ESBL producing *E. coli* and *K. pneumoniae* isolates.**

| Antimicrobial Agents | ESBL producing | | | Non -ESBL producing | | Total (N = 263) |
|---|---|---|---|---|---|---|
| | *E. coli* (N = 39) | *K. pneumonia* (N = 8) | Total | *E. coli* (N = 185) | *K. pneumonia* (N = 31) | |
| | Resistance | Resistance | | Resistance | Resistance | Resistance |
| Ampicillin | 37(94.9) | 8(100) | 45(95.7) | 127(68.6) | 31(100) | 203(77.2) |
| Amoxicillin/Clavulanic Acid | 17(43.6) | 2(25) | 19(40.4) | 50 (27.0) | 3 (9.7) | 72(27.3) |
| Ampicillin/Sulbactam | 30(76.9) | 6(75.5) | 36(76.5) | 101 (54.6) | 10(32.3) | 147(55.8) |
| Cefazolin | 29(74.4) | 7(87.5) | 36(76.6) | 23 (12.4) | 1 (3.2) | 60(22.8) |
| Cefuroxime | 34(87.2) | 7(87.5) | 41(87.2) | 16 (8.6) | 0(0) | 57(21.7) |
| Cefuroxime Axetil | 37(94.9) | 7(87.5) | 44(93.6) | 22 (11.9) | 1 (3.2) | 67(25.7) |
| Ceftazidime | 28(71.8) | 7(87.5) | 35(74.5) | 0 (0) | 0(0) | 35(13.3) |
| Ceftriaxone | 29(74.4) | 7(87.5) | 36(76.6) | 0(0) | 0(0) | 36(13.7) |
| Cefepime | 29(74.4) | 7(87.5) | 36(76.6) | 0(0) | 0(0) | 36(13.7) |
| Ertapenem | 0(0.0) | 0(0.0) | 0(0.0) | 0(0) | 0(0) | 0(0) |
| Imipenem | 0(0.0) | 0(0.0) | 0(0.0) | 0(0) | 0(0) | 0(0) |
| Gentamicin | 4(10.3) | 4(50.0) | 8(17.0) | 1 (0.5) | 1 (3.2) | 10(3.8) |
| Tobramycin | 7(18.0) | 5(62.5) | 12(25.5) | 2 (1.0) | 1 (3.2) | 15(5.7) |
| Ciprofloxacin | 10(25.6) | 2(25.0) | 12 (25.5) | 3 (1.5) | 0(0) | 15(5.7) |
| Levofloxacin | 9(23.1) | 1(12.5) | 10(21.3) | 3 (1.6) | 0(0) | 13(4.9) |
| Tetracycline | 29(74.4) | 4(50.0) | 33(70.2) | 105(56.7) | 13 (41.9) | 151(57.4) |
| Nitrofurantoin | 8(20.5) | 4(50.0) | 12(25.5) | 7(3.7) | 1 1(35.5) | 30(11.4) |
| Trimethoprim/Sulfemethoxazole | 32(82.1) | 5(62.5) | 37(78.7) | 110(59.5) | 13(41.9) | 160(60.8) |

*E. coli* and *K. pneumoniae*. In the present study, ESBL and non-ESBL producing E. *coli* and K. *pneumoniae* isolates were 100% susceptible to carbapenems (ertapenem and imipenem) which are the most active drugs against ESBL producing E. *coli* and K. *pneumoniae* isolates (Table 4).

## Multi-drug resistant pattern

Overall, 65.4% (172/263) of ESBL producing and non-ESBL producing *E. coli* and *K. pneumoniae* isolates were multidrug-resistant (MDR, resistance to at least 3 antibiotics from different classes). Among the total MDR, *E. coli* was 86.6% (149/172) and *K. pneumonia* was 13.4% (23/172). The overall proportion of MDR was 93.6% among ESBL producers and 59% among non-ESBL producers of *E. coli* and *K. pneumoniae* isolates. ESBL producers of *E. coli* accounted for 94.9% (37/39) and *K. pneumoniae* accounted for 87.5% (7/8) of MDR. Non-ESBL producing *E. coli* and *K. pneumonia* accounted for 60.5% *(*112/185) and 51.6% (16/31) MDR respectively (Table 5).

## Risk factors associated with ESBL fecal carriage

In this study, different types of possible risk factors were analyzed and only children's mothers who had lower educational level (primary school) (OR: 2.472, 95% CI: 1.323–4.618, P = 0.0062) and children who used tap water for drinking (OR: 1.714, 95% CI: 1.001–3.659, P = 0.048) were found to be significantly associated with higher ESBL fecal carriage. On the other hand, there was no significant association were found among age groups (OR: 1.379, 95% CI: 0.791–2.406, P = 0.252), sex (OR: 1.069, 95% CI: 0.632–1.810, P = 0.803), previous antibiotics usage (OR: 0.883, 95% CI: 0.469–1.663, P = 0.703), hospital visit (OR: 0.806, 95% CI: 0.383–1.693, P = 0.563), hospital admission (OR: 0.824, 95% CI: 0.279–2.432, P = 0.721), and undergoing a surgical operation in the last 12 months (OR: 1.208, 95% CI: 1.144–1.276, P = 0.519) with higher ESBL fecal carriage (Table 6).

**Table 5. Multidrug resistance pattern of ESBL producing and non-ESBL producing *E. coli* and *K. pneumoniae* isolates.**

| | Isolates | Level of antibiotics resistance ((number (%)) | | | | | | | | MDR ($\geq$R3) |
|---|---|---|---|---|---|---|---|---|---|---|
| | | $R_0$ | $R_1$ | $R_2$ | $R_3$ | $R_4$ | $R_5$ | $R_6$ | $\geq R_7$ | |
| ESBL positive | *E. coli* (n = 39) | 0 (0.0) | 1 (2.6) | 1 (2.6) | 3 (7.7) | 3 (7.7) | 10 (25.6) | 10 (25.6) | 11 (28.2) | 37 (94.9) |
| | *K. pneumonia* (n = 8) | 0 (0.0) | 0 (0.0) | 1 (12.5) | 1 (12.5) | 0 (0.0) | 1 (12.5) | 0 (0.0) | 5 (62.5) | 7 (87.5) |
| ESBL negative | *E. coli* (n = 185) | 42 (22.7) | 11 (5.9) | 20 (10.8) | 26 (14.1) | 62 (33.5) | 22 (11.9) | 1 (0.5) | 1 (0.5) | 112 (60.5) |
| | *K. pneumonia* (n = 31) | 0 (0.0) | 9 (29.0) | 6 (19.4) | 6 (19.4) | 8 (25.8) | 2 (6.5) | 0 (0.0) | 0 (0.0) | 16 (51.6) |
| Total (N = 263) | | 42 (16.0) | 21 (8.0) | 28 (10.6) | 36 (13.7) | 73 (27.8) | 35 (13.3) | 11 (4.2) | 17 (6.5) | **172 (65.4)** |

R0: resistance to no antibiotics, R1-6: resistance to 1, 2, 3, 4, 5, and 6, $\geq$R7: resistance to $\geq$7 antibiotics; MDR: Multidrug resistance.

MDR ($\geq$R3): resistance to 3 or more antibiotics from different classes.

**Table 6. Risk factors for ESBL fecal carriage among under five years children.**

| Characteristics | | | Total | ESBL | | OR | 95%CI | | $X^2$ | p-value |
|---|---|---|---|---|---|---|---|---|---|---|
| | | | N = 269 | Negative N = 223 | Positive N = 46 | | Lower | Upper | | |
| | | | | N (%) | N (%) | | | | | |
| Age | 29days-23months | | 155 | 125(80.6) | 30(19.4) | 1.379 | 0.791 | 2.406 | 1.311 | 0.252 |
| | 24–59 months | | 114 | 98(86.0) | 16(14.0) | | | | | |
| Sex | Male | | 130 | 107(82.3) | 23(17.7) | 1.069 | 0.632 | 1.810 | 0.62 | 0.803 |
| | Female | | 139 | 116(83.5) | 23(16.5) | | | | | |
| Prior intake of antibiotics | Yes | | 216 | 180(83.3) | 36(16.7) | 0.883 | 0.469 | 1.663 | 0.145 | 0.703 |
| | No | | 53 | 43(81.1) | 10(18.9) | | | | | |
| Hospital visit | Yes | | 49 | 42(85.7) | 7(14.3) | 0.806 | 0.383 | 1.693 | 0.335 | 0.563 |
| | No | | 220 | 181(82.3) | 39(17.7) | | | | | |
| Previous hospital admission | Yes | | 21 | 18(85.7) | 3(14.3) | 0.824 | 0.279 | 2.432 | 0.127 | 0.721 |
| | No | | 248 | 205(82.7) | 43(17.3) | | | | | |
| Previous surgery | Yes | | 2 | 2(100.0) | 0(0.0) | 1.208 | 1.144 | 1.276 | 0.416 | 0.519 |
| | No | | 267 | 221(82.8) | 46(17.2) | | | | | |
| Mother Educational level | Illiterate/cannot read and write/ | | 53 | 46(86.8) | 7(13.2) | 0.690 | 0.2901 | 1.644 | 0.405 | 0.5245 |
| | Illiterate /able to read and write/ | | 3 | 3(100.0) | 0(0.0) | 0.797 | 0.0392 | 1.618 | 0.318 | 0.5727 |
| | Primary | | 130 | 105(80.8) | 25(19.2) | 2.472 | 1.323 | 4.618 | 7.487 | 0.0062 |
| | Secondary | | 61 | 50(82.0) | 11(18.0) | 1.087 | 0.5152 | 2.295 | 0.000 | 0.9788 |
| | College Graduate | | 10 | 8(80.0) | 2(20.0) | 1.222 | 0.2509 | 5.948 | 0.032 | 0.8573 |
| | Don't know | | 12 | 11(91.7) | 1(8.3) | 0.428 | 0.0539 | 3.401 | 0.187 | 0.6650 |
| Source of drinking water for the child | Tap Water | Yes | 128 | 100(78.1) | 28(21.9) | 1.714 | 1.001 | 3.659 | 3.927 | 0.048 |
| | | No | 141 | 123(87.2) | 18(12.8) | | | | | |
| | Boiled & Cooled Water | Yes | 26 | 23(88.5) | 3(11.5) | 0.652 | 0.217 | 1.956 | 0.687 | 0.407 |
| | | No | 243 | 200(82.3) | 43(17.7) | | | | | |
| | Treated Water | Yes | 34 | 31(91.2) | 3(8.8) | 0.482 | 0.158 | 1.469 | 1.881 | 0.178 |
| | | No | 235 | 192(81.7) | 43(18.3) | | | | | |
| | Bottled Water | Yes | 113 | 94(83.2) | 19(16.8) | 0.971 | 0.569 | 1.658 | 0.11 | 0.915 |
| | | No | 156 | 129(82.7) | 27(17.3) | | | | | |
| | Filtered Water | Yes | 1 | 1(100.0) | 0(0.0) | 1.207 | 1.143 | 1.275 | 0.376 | 0.540 |
| | | No | 268 | 222(82.8) | 46(17.2) | | | | | |

## Discussion

Different studies were conducted on the prevalence of ESBL fecal carriage around the world and showed a large variation from country to country. There were limited data on the fecal carriage of ESBL producing *E. coli* and *K. pneumoniae* and to the best of our knowledge, this is the first study that reported the prevalence of ESBL fecal carriage among children under five years in Ethiopia.

In the present study, the overall prevalence of ESBL-producing *E. coli* and *K. pneumoniae* fecal carriage among children under five years was 17.1% (46/269; 95% CI: 12.9%–22.7%). This prevalence is lower compared to the report of a previous study by Desta K et al. [34], which reported that 52% of hospitalized patients were carriers of ESBL. It is also lower than the study in Guinea-Bissau (32.6%) [13] and Tanzania (34.3%) [35]. In contrast, the prevalence in this study is relatively comparable with the study in the Lao People's Democratic Republic (23.2%) [36], Spain (24.0%) [37], and Lebanese (24.8%) [12]. On the other hand, the prevalence of ESBL observed in this study is higher compared to the study in Madagascar (10.1%) [17], South Africa (4.7%) [7], and France (4.6%) [21]. The differences seen in the prevalence might be due to the differences in study participants, study settings, and countries.

In the present study, the predominant ESBL producing isolate was *E. coli* (83%) followed by *K. pneumoniae* (17%). Similarly, *E. coli* was the predominant ESBL-producing isolate in a study from Turkey [38] and Tanzania [5]. The studies in Lao People's Democratic Republic [36] and Zimbabwe [39] also showed that 83.3%, 78%, and 95.8% ESBL producing isolate was *E. coli*, respectively. Studies in Madagascar [17], Korea [40], and Lebanese [12] also demonstrated that *E. coli* is the predominant ESBL-producing isolate.

In this study, ESBL producing *E. coli* and K. *pneumonia* isolates showed a high level of resistance against trimethoprim/sulfamethoxazole (78.7%), followed by tetracycline (70.2%), ciprofloxacin (25.5%), and gentamicin (17%). Similarly, a study from Tanzania was reported high resistance to tetracycline (100%), trimethoprim-sulfamethoxazole (97%), ciprofloxacin (69%), and gentamicin (44%) [5]. In Madagascar, trimethoprim-sulfamethoxazole (91.3%), gentamicin (76.1%), and ciprofloxacin (50.0%) were resistant [41]. In Spain, Nalidixic acid (64.7%), ciprofloxacin (32.4%), levofloxacin (32.4%), and trimethoprim-sulfamethoxazole (41.2%) were resistant [37].

In our study, the overall prevalence of MDR was 93.6% among ESBL producer *E. coli* and *K. pneumoniae* isolates. This result is in line with the findings of studies conducted in Tanzania [35], MDR was 94%, in Guinea-Bissau [13], nearly all isolates were MDR and in Madagascar [41], most ESBL isolates were MDR strains. However, it was higher than a study conducted in Spain, 52.4% of the ESBL were resistant to three or more antimicrobial classes [12].

ESBL-producing bacteria are frequently associated with co-resistance to non-beta-lactam antimicrobial agents as demonstrated in several studies [5, 12, 13, 17], which may critically complicate the treatment of severe bacterial infections and leaves very few choices for treatment (limits the therapeutic choice to carbapenems). Interestingly, unlike the previous study by Desta et al. [34], which reported that 2% (5/267) of ESBL producing isolates were resistant to carbapenem and all detected in children, in the present study all ESBL producing *E. coli* and *K. pneumoniae* isolates were susceptible to carbapenems (ertapenem and imipenem), which are the most active drugs against ESBL producer.

The risk factors analysis data in this study demonstrated that children's mothers who had lower educational level (primary school) (OR: 2.472, 95% CI: 1.323–4.618, P = 0.0062) and children who used tap water for drinking (OR: 1.714, 95% CI: 1.001–3.659, P = 0.048) were found to be significantly associated with higher ESBL fecal carriage. Several studies reported the presence of ESBL producing isolates in drinking water [42–44], which could serve as reservoirs for ESBL producing bacteria.

On the other hand, the finding from this study revealed that there was no association between age and ESBL carriage (OR: 1.379, 95% CI: 0.791–2.406, p = 0.252), which is similar to a study in Guinea-Bissau [13] and Madagascar [17]. However, this result is different from the study by Tellevik MG et al. [35] which reported that having age equal to or below 12 months was significantly associated with ESBL carriage (P = 0.012; OR = 1.82; 95% CI: 1.14–2.91), this was also different from French study [21] in which the risk of ESBL carriage was higher among children over 1 year old than in younger children (6.5% versus 2.5%, respectively; OR = 2.69, 95% CI [0.95–7.61]). Our study also showed that there was no significant difference between male and female ESBL carriers (OR: 1.069, 95% CI: 0.632–1.810, P = 0.801), this finding is similar to a study by Herindrainy P et al. [17] and Erdoğan DC et al. [38] but different from the study by Hijazi SM et al. [12], which reported that males had a higher colonization frequency (33.9%) than did females (15.9%) (P = 0.09).

The risk analysis result of the present study also showed that there was no association between the previous usage of antibiotics (in the last 12 months) and ESBL colonization (OR: 0.883, 95% CI: 0.469–1.667, P = 0.703), which is in agreement with the study conducted in Guinea-Bissau [13], Tanzania [5] and Lebanese [12]. However, it is different from other studies in Tanzania [35], the Democratic Republic [36], and France [21] that indicated the use of antibiotics as a risk factor for ESBL carriage.

In Turkey [38] and Lebanese [12] studies, ESBL carriage rates were found to be significantly higher in those who were hospitalized. However, this study was not demonstrated a significant association between previous hospital admission and high rate of ESBL carriage (OR: 0.824, 95% CI: 0.279–2.432, P = 0.721), this is similar to the study in Guinea-Bissau [13]. The study by Erdoğan DC et al. [38] also noted that undergoing a surgical operation (P = 0.005) was associated with higher ESBL carriage rates. However, our finding not showed an association between ESBL carriage and undergoing a surgical operation. In this study, hospital visits (OR: 0.806, 95% CI: 0.383–1.693, P = 0.563) in the last 12 months were also not associated with higher ESBL carriage rates.

## Limitation

This study was limited to test ESBL production only in *E. coli* and *K. pneumoniae*. In addition, a negative ESBL test result does not rule out the presence of an ESBL masked by an AmpC beta-lactamase. Therefore, this may probably lower the prevalence. Moreover, molecular characterization of ESBL encoding genes was not conducted due to our laboratory has no facility for molecular analysis.

## Conclusions

This study was reported a high prevalence rate (17.1%) of ESBL producing *E. coli* and *K. pneumoniae* fecal carriage among children under five years. ESBL producing isolates also showed high levels of MDR (93.6%) and high rates of co-resistance to aminoglycosides, fluoroquinolones, and trimethoprim-sulfamethoxazole. This suggested that the necessity of routine screening of ESBL is crucial for the early detection and appropriate antibiotics selection for infection caused by ESBL producing pathogens. The risk analysis of this study demonstrated that children's mothers who had lower educational levels (primary school) and children who used tap water for drinking were found to be significantly associated with higher ESBL fecal carriage.

## Acknowledgments

We are very grateful to Ethiopian Public Health Institute Clinical Bacteriology and Mycology National Reference Laboratory for allowing us to do this study in their laboratory. We would

also like to thank all study subjects who were volunteered to participate in this study and Addis Raey health center staff who assisted with the study subject's recruitment.

## Author Contributions

**Conceptualization:** Mekdes Alemu Tola, Surafel Fentaw Dinku, Kassu Desta Tullu.

**Data curation:** Mekdes Alemu Tola, Surafel Fentaw Dinku.

**Formal analysis:** Mekdes Alemu Tola, Negga Asamene Abera, Yonas Mekonnen Gebeyehu.

**Funding acquisition:** Mekdes Alemu Tola.

**Investigation:** Mekdes Alemu Tola, Negga Asamene Abera, Yonas Mekonnen Gebeyehu, Surafel Fentaw Dinku.

**Methodology:** Mekdes Alemu Tola.

**Project administration:** Mekdes Alemu Tola, Kassu Desta Tullu.

**Resources:** Mekdes Alemu Tola, Negga Asamene Abera, Yonas Mekonnen Gebeyehu.

**Software:** Mekdes Alemu Tola.

**Supervision:** Mekdes Alemu Tola, Surafel Fentaw Dinku, Kassu Desta Tullu.

**Validation:** Mekdes Alemu Tola.

**Visualization:** Mekdes Alemu Tola.

**Writing – original draft:** Mekdes Alemu Tola.

**Writing – review & editing:** Mekdes Alemu Tola, Negga Asamene Abera, Yonas Mekonnen Gebeyehu, Surafel Fentaw Dinku, Kassu Desta Tullu.

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
