## [Decision Letter · Decision Letter 0]

25 Aug 2021

PONE-D-21-25316

Fecal carriage of extended spectrum beta -lactamase producing Escherichia coli and Klebsiella pneumoniae among children under five years in Addis Ababa, Ethiopia.   

PLOS ONE

Dear Dr. Tola,

Thank you for submitting your manuscript to PLOS ONE. After careful consideration, we feel that it has merit but does not fully meet PLOS ONE’s publication criteria as it currently stands. Therefore, we invite you to submit a revised version of the manuscript that addresses the points raised during the review process.

 A major revision is required. The manuscript should be revised for English Editing and grammar mistakes. 

We look forward to receiving your revised manuscript.

Kind regards,

Abdelazeem Mohamed Algammal, Prof, Ph.D

Academic Editor

PLOS ONE

Journal Requirements:

2. Please amend your current ethics statement to address the following concern:

Please explain i) why written consent was not obtained, ii) how you documented participant consent, and iii) whether the ethics committees/IRB approved this consent procedure.

Reviewers' comments:

Reviewer's Responses to Questions

**Comments to the Author**

1. Is the manuscript technically sound, and do the data support the conclusions?

Reviewer #1: Yes

Reviewer #2: Yes

2. Has the statistical analysis been performed appropriately and rigorously? 

Reviewer #1: N/A

Reviewer #2: Yes

3. Have the authors made all data underlying the findings in their manuscript fully available?

Reviewer #1: Yes

Reviewer #2: Yes

4. Is the manuscript presented in an intelligible fashion and written in standard English?

Reviewer #1: No

Reviewer #2: Yes

5. Review Comments to the Author

Reviewer #1: Comments to authors:

- The current study is interesting; however, the authors should address the following comments to improve the quality of the manuscript:

- The manuscript should be revised for language editing and grammar mistakes by a native English speaker.

Title:

I think the work would benefit from the title that contains the main conclusion of the study (should be derived from the conclusion). Please modify the title.

Abstract:

- The abstract must illustrate the used methods and the most prevalent results (give more hints about methods and results). Besides, rephrase the main conclusion of your findings.

Introduction:

-Give a hint about different infections caused by E. coli and K. pneumonia, their virulence factors, and the mechanism of disease occurrence.

- The authors should illustrate the public health importance concerning the emergence of multidrug-resistant (MDR) bacterial pathogens that reflecting the necessity of new potent and safe antimicrobial agents. Several studies proved the widespread MDR- bacterial pathogens;

Authors could add the following paragraph:

Multidrug resistance has been increased all over the world that is considered a public health threat. Several recent investigations reported the emergence of multidrug-resistant bacterial pathogens from different origins including humans, poultry, cattle, and fish that increase the need for routine application of the antimicrobial susceptibility testing to detect the antibiotic of choice as well as the screening of the emerging MDR strains. You should cite the following valuable studies:

PMID: 33177849

1-PMID: 32497922

2-PMID:33061472

3-PMID: 33947875

4-PMID: 32472209

5-PMID: 32994450

6-PMID: 33188216

7-PMID: 32235800

-Rephrase the aim of the work to be clear and better sound.

Material and methods

- Bacterial isolation and identification:

•Explain in detail the methods of the bacterial isolation and identification (add specific references, the used media: add the company and country, the used biochemical reactions). The authors are advised to perform the conventional methods of bacterial isolation and identification as well as the Vitek-2. Besides, add more details about the Vitek identification system.

- Antimicrobial susceptibility testing:

•Illustrate the antimicrobial classes of the tested antimicrobial agents.

- PCR-based detection of the Extended-spectrum beta-lactamase genes (such as blaTEM, blaCTX-M, blaSHV, and bla-KPC…etc.) in the recovered isolated should be carried out for necessary.

- Add more details about the used program (SPSS) in the statistical analyses.

-Results:

-Illustrate the statistical analyses of the results presented in Tables 1-5.

-The subtitle: Bacteria isolates should be replaced by: Prevalence of E. coli and Klebsiella spp.

-Improve the presentation of your finding using illustrating figures.

-PCR-based detection of the Extended-spectrum beta-lactamase genes (such as blaTEM, blaCTX-M, blaSHV, and bla-KPC…etc.) in the recovered isolated should be carried out for necessary. Besides, the authors should add the PCR figures.

-Discussion:

- The authors are advised to illustrate the real impact of their findings without repetition of results.

-Remove all subtitles from the discussion section.

-Conclusion

- Should be rephrased to be sounded. A real conclusion should focus on the question or claim you articulated in your study, which resolution has been the main objective of your paper?

Reviewer #2: - The current study has a significant impact, but it needs a major revision:

- The manuscript should be revised for grammar mistakes.

- Please write the scientific names of all pathogens in italic form all over the manuscript.

-The title is broad, please modify the title.

- Add more details about the used methods and most prevalent results in the abstract.

-In the introduction: discuss the public health importance of the recovered bacterial pathogens and different infection caused by them.

-Improve the aim of work.

Methods:

-Discuss in details the methods of isolation and identification of the bacterial pathogens (by traditional methods and Vitek).

-Specific references should be added to all the used methods and techniques.

-Add the manufacturing company, city, and country for the used media and antimicrobial discs.

- The Extended spectrum beta-lactamase genes should be genetically detected using PCR.

-Add more details about the used software in the statistical analysis.

-Results:

- The Extended spectrum beta-lactamase genes should be genetically detected using PCR (support your findings with the PCR Figures).

-Where are the statistical analyses? (only used in Table 6).

-Discussion:

- Please improve. Delete all the subtitles.

-Please improve the main conclusion of the manuscript.

6. PLOS authors have the option to publish the peer review history of their article (what does this mean?). If published, this will include your full peer review and any attached files.

Reviewer #1: No

Reviewer #2: No

---

## [Author Response · Author response to Decision Letter 0]

16 Sep 2021

Responses to the Comments from Reviewer #1

Comments: The current study is interesting; however, the authors should address the following comments to improve the quality of the manuscript: The manuscript should be revised for language editing and grammar mistakes by a native English speaker.

Response: Thank you for the comment and we have incorporated all your comments and suggestion in the revised manuscript accordingly. The manuscript is also reviewed for the English language. 

Title

Comments: I think the work would benefit from the title that contains the main conclusion of the study (should be derived from the conclusion). Please modify the title.

Response: Thank you for this suggestion and the title is modified to “High prevalence of extended-spectrum beta-lactamase-producing Escherichia coli and Klebsiella pneumoniae fecal carriage among children under five years in Addis Ababa, Ethiopia”. 

Abstract

Comments: The abstract must illustrate the used methods and the most prevalent results (give more hints about methods and results). Besides, rephrase the main conclusion of your findings.

Response: Thank you for the comment. We have made revisions accordingly.

Introduction

Comments: Give a hint about different infections caused by E. coli and K. pneumonia, their virulence factors, and the mechanism of disease occurrence. The authors should illustrate the public health importance concerning the emergence of multidrug-resistant (MDR) bacterial pathogens that reflecting the necessity of new potent and safe antimicrobial agents. 

Several studies proved the widespread MDR- bacterial pathogens; Authors could add the following paragraph: 

Multidrug resistance has been increased all over the world that is considered a public health threat. Several recent investigations reported the emergence of multidrug-resistant bacterial pathogens from different origins including humans, poultry, cattle, and fish that increase the need for routine application of the antimicrobial susceptibility testing to detect the antibiotic of choice as well as the screening of the emerging MDR strains. You should cite the following valuable studies: 

PMID: 33177849

1-PMID: 32497922

2-PMID:33061472

3-PMID: 33947875

4-PMID: 32472209

5-PMID: 32994450

6-PMID: 33188216

7-PMID: 32235800

-Rephrase the aim of the work to be clear and better sound. 

Response: Thank you for the valuable comment and included in the revised manuscript.

Material and methods 

Comments: Bacterial isolation and identification: Explain in detail the methods of bacterial isolation and identification (add specific references, the used media: add the company and country, the used biochemical reactions). The authors are advised to perform the conventional methods of bacterial isolation and identification as well as the Vitek-2. Besides, add more details about the Vitek identification system. 

Antimicrobial susceptibility testing: Illustrate the antimicrobial classes of the tested antimicrobial agents. 

PCR-based detection of the Extended-spectrum beta-lactamase genes (such as blaTEM, blaCTX-M, blaSHV, and bla-KPC…etc.) in the recovered isolated should be carried out for necessary.

Add more details about the used program (SPSS) in the statistical analyses. 

Response: Thank you for this suggestion. We have made revisions for all comments accordingly. Concerning PCR-based detection of ESBL genes, it would have been interesting if we could perform it. However, the molecular characterization of ESBL encoding genes was not conducted due to our laboratory ( even though it is a national reference lab) has no facility for molecular analysis. Our study only detected ESBL production phenotypically and is mentioned under limitation. 

Results

Comments: Illustrate the statistical analyses of the results presented in Tables 1-5. 

-The subtitle: Bacteria isolates should be replaced by: Prevalence of E. coli and Klebsiella spp. 

-Improve the presentation of your finding using illustrating figures. 

-PCR-based detection of the Extended-spectrum beta-lactamase genes (such as blaTEM, blaCTX-M, blaSHV, and bla-KPC…etc.) in the recovered isolated should be carried out for necessary. Besides, the authors should add the PCR figures. 

Response: Thank you for the comment. We have made revisions for all comments accordingly. Concerning presenting the finding by figures, the figure is better to interpreted data easily than a table. However, in our case, we found that it is much easier to put our data in tables than figures. The PCR-based detection of ESBL genes was not performed due to our laboratory has no facility for molecular analysis we have only done phenotypic detection of ESBL and this is mentioned under limitation. 

Discussion

Comments: The authors are advised to illustrate the real impact of their findings without repetition of results. Remove all subtitles from the discussion section.

Response: Thank you for the comment. The subtitles are removed and the discussion is improved in the revised manuscript.

Conclusion 

Comments: Should be rephrased to be sounded. A real conclusion should focus on the question or claim you articulated in your study, which resolution has been the main objective of your paper?

Response: Thank you for the comment and is improved in the revised manuscript.

Responses to the Comments from Reviewer #2

Comments: The current study has a significant impact, but it needs a major revision: The manuscript should be revised for grammar mistakes. Please write the scientific names of all pathogens in italic form all over the manuscript. 

Response: Thank you for the comment and we have incorporated all your comments and suggestion in the revised manuscript accordingly. The manuscript is also reviewed for the English language. In addition, the scientific names of all pathogens have been written in italic form all over the revised manuscript. 

Title

Comments: The title is broad, please modify the title.

Response: Thank you for this suggestion and the title is modified to “High prevalence of extended-spectrum beta-lactamase-producing Escherichia coli and Klebsiella pneumoniae fecal carriage among children under five years in Addis Ababa, Ethiopia”. 

Abstract

Comments: Add more details about the used methods and most prevalent results in the abstract. 

Response: Thank you for the comment. We have made revisions accordingly.

Introduction

Comments: In the introduction: discuss the public health importance of the recovered bacterial pathogens and different infections caused by them. Improve the aim of work.

Response: Thank you for the valuable comment and is included in the revised manuscript.

Material and methods 

Comments: Discuss in detail the methods of isolation and identification of the bacterial pathogens (by traditional methods and Vitek). 

- Specific references should be added to all the used methods and techniques. 

-Add the manufacturing company, city, and country for the used media and antimicrobial discs. 

-The Extended-spectrum beta-lactamase genes should be genetically detected using PCR. 

-Add more details about the used software in the statistical analysis. 

Response: Thank you for this suggestion. We have made revisions for all comments accordingly. Concerning PCR-based detection of ESBL genes, it would have been interesting if we could perform it. However, the molecular characterization of ESBL encoding genes was not conducted due to our laboratory ( even though it is a national reference lab) has no facility for molecular analysis. Our study only detected ESBL production phenotypically and is mentioned under limitation.

Results

Comments: The Extended-spectrum beta-lactamase genes should be genetically detected using PCR (support your findings with the PCR Figures). 

-Where are the statistical analyses? (Only used in Table 6). 

Response: Thank you for the comment. Our study only detected ESBL production phenotypically due to our laboratory has no facility for molecular analysis and is mentioned under limitation. 

We used simple frequency to describe the study population's socio-demographic, clinical condition, and prevalence of ESBL fecal carriage. Besides, we tried to explore the possible risk factors (like antibiotic usage, hospital visit, and admission, previous surgery) associated with ESBL fecal carriage (we got this information from previous studies) and tried to look in our study and summarized in Table 6. 

Discussion

Comment:s Please improve. Delete all the subtitles.

Response: Thank you for the comment. The subtitles are removed and the discussion is improved in the revised manuscript.

Conclusion 

Comment: Please improve the main conclusion of the manuscript.

Response: Thank you for the comment and is improved in the revised manuscript.

Additional clarifications

Comment: Please amend your current ethics statement to address the following concern:

Please explain 

i) why written consent was not obtained, 

ii) how you documented participant consent, and 

iii) whether the ethics committees/IRB approved this consent procedure.

Response: Thank you for the comment. The ethics statement has been amended in regards to participant consent from verbal to written informed consent in the revised manuscript. It was mistakenly written as verbal informed consent. 

In addition to the above comments, all spelling and grammatical errors pointed out by the reviewers have been corrected.

---

## [Decision Letter · Decision Letter 1]

20 Sep 2021

High prevalence of  extended spectrum beta -lactamase producing Escherichia coli and Klebsiella pneumoniae fecal carriage among children under five years in Addis Ababa, Ethiopia.

PONE-D-21-25316R1

Dear Dr. Tola,

We’re pleased to inform you that your manuscript has been judged scientifically suitable for publication and will be formally accepted for publication once it meets all outstanding technical requirements.

Kind regards,

Abdelazeem Mohamed Algammal, Prof, Ph.D

Academic Editor

PLOS ONE

Additional Editor Comments (optional):

Reviewers' comments:

Reviewer's Responses to Questions

**Comments to the Author**

1. If the authors have adequately addressed your comments raised in a previous round of review and you feel that this manuscript is now acceptable for publication, you may indicate that here to bypass the “Comments to the Author” section, enter your conflict of interest statement in the “Confidential to Editor” section, and submit your "Accept" recommendation.

Reviewer #1: All comments have been addressed

2. Is the manuscript technically sound, and do the data support the conclusions?

Reviewer #1: Yes

3. Has the statistical analysis been performed appropriately and rigorously? 

Reviewer #1: Yes

4. Have the authors made all data underlying the findings in their manuscript fully available?

Reviewer #1: Yes

5. Is the manuscript presented in an intelligible fashion and written in standard English?

Reviewer #1: Yes

6. Review Comments to the Author

Reviewer #1: The authors have carried out a significant changes to the manuscript. They have addressed all the suggested corrections and comments. Really, it's an interesting study that has a significant impact. Now, the manuscript could be accepted.

Congratulations.

7. PLOS authors have the option to publish the peer review history of their article (what does this mean?). If published, this will include your full peer review and any attached files.

Reviewer #1: No

---

## [Editor Report · Acceptance letter]

24 Sep 2021

PONE-D-21-25316R1 

*High prevalence of extended-spectrum beta-lactamase-producing Escherichia coli and Klebsiella pneumoniae fecal carriage among children under five years in Addis Ababa, Ethiopia.*

Dear Dr. Tola:

I'm pleased to inform you that your manuscript has been deemed suitable for publication in PLOS ONE. Congratulations! Your manuscript is now with our production department. 

Kind regards, 

on behalf of

Professor Abdelazeem Mohamed Algammal 

Academic Editor

PLOS ONE